# Health Risk Assessment of Groundwater Contaminated by Oil Pollutants Based on Numerical Modeling

**DOI:** 10.3390/ijerph16183245

**Published:** 2019-09-04

**Authors:** Xue Bai, Kai Song, Jian Liu, Adam Khalifa Mohamed, Chenya Mou, Dan Liu

**Affiliations:** Faculty of Geosciences and Environmental Engineering, Southwest Jiaotong University, Chengdu 610031, China (X.B.) (J.L.) (A.K.M.) (C.M.) (D.L.)

**Keywords:** carcinogenic risk, grey relational analysis, health risk assessment, non-carcinogenic risk

## Abstract

To provide theoretical support for the protection of dispersed drinking water sources of groundwater, we need to accurately evaluate the time and scope of groundwater pollution hazards to human health. This helps the decision-making process for remediation of polluted soil and groundwater in service stations. In this study, we conducted such an evaluation by coupling numerical modeling with a health risk assessment. During the research, soil and groundwater samples were collected and analyzed for 20 pollutants. Fifty-six percent of the heavy contaminants and 100% of the organic contaminants exhibited maximum values at the location of the oil depot. Gray correlation analysis showed that the correlation between background samples and soil underlying the depot was 0.375–0.567 (barely significant to insignificant). The correlation between the reference sequence of other points was 0.950–0.990 (excellent correlation). The correlation of environmental impact after oil depot leakage followed the order: organic pollutants > heavy metals > inorganic pollutants. The groundwater simulation status and predictions indicated that non-carcinogenic health risks covered 25,462 m^2^ at the time of investigation, and were predicted to extend to 29,593 m^2^ after five years and to 39,873 m^2^ after 10 years. Carcinogenic health risks covered 21,390 m^2^ at the time of investigation, and were predicted to extend to 40,093 m^2^ after five years and to 53,488 m^2^ after 10 years. This study provides theoretical support for the protection of a dispersed drinking water source such as groundwater, and also helps the decision-making process for groundwater and soil environment improvement.

## 1. Introduction

Vehicle service stations and underground oil depots are among the main polluting sites of soil and groundwater because of the leakage of oil storage tanks and oil transshipment spills [1,2,3]. When an oil tank leaks, oil will pollute the surrounding groundwater and soil. As a non-aqueous liquid, oil will be first intercepted in the soil pore [4], which will not only change the soil structure and lead to settlement, but also become a potential source of contamination for groundwater. Over time, even pollutants trapped in the vadose zone will migrate vertically into an aquifer under the effect of leaching [5,6,7,8]. According to previous research, the composition of stored oil products at service stations or underground oil tanks is complex, and controlled by production processes and standard requirements [9,10,11]. Adeniji et al. [12] studied the types and composition of oil pollutants; Innocenti et al. [13], Lueders et al. [14], Wade et al. [15], and Meyer et al. [16] studied the main organic pollutants in oil, including methyl tert-butyl ether, benzene series, total petroleum hydrocarbons, and polycyclic aromatic hydrocarbons, as well as their migration in the environment. After entering the ecological environment, pollutants contained in oil will affect the growth and development of animals and plants. Some refractory components can be enriched in humans and animals through drinking water and diet and even cause cancer [17,18,19,20].

The environmental risk of oil tank leakage increases as the tanks age. According to the US Environmental Protection Agency’s (USEPA) statistics for 2011, 71% of the buried oil tanks in the USA that have operated for more than 10 years have leaked [21]. As of 2015, the majority of service stations in China still used single-wall tanks [22]. An investigation and monitoring of soil and groundwater at 242 service stations in six provinces of China showed that the proportion of pollutants detected at the stations in each province was 24–89% [23]. In the red beds regions of southwestern China, many rural residents still extract drinking water directly from groundwater that has not been purified or treated. If such drinking water sources are close to a source of pollution, human health can be harmed through drinking contaminated water. Therefore, research on the prevention and control of groundwater pollution cannot be ignored.

Numerical models are the main method by which to study and solve many environmental problems. Models that predict groundwater dynamics can be used to calculate the migration path and concentration trend of pollutants in the environment by generalizing hydrogeological conditions and designing various working conditions; such models have been used widely [24,25]. Environmental risk assessment reflects the change of environmental protection from post-pollution treatment to pre-pollution prediction and management. The “four step method” proposed by the US National Academy of Sciences consists of hazard identification, dose-effect relationship assessment, exposure assessment and risk characterization. Based on this “four step method”, the USEPA proposed a method for assessing the risk of carcinogenesis from toxic chemicals. The USEPA method has been recognized and widely used to assess the risk of human health at contaminated sites. Zhang et al. [26], Ching-Ping et al. [27], and Sany et al. [28] applied the USEPA method to assess the health risks of various pollutants from different sources. A numerical risk assessment value is actually a function related to a pollutant concentration, and most of the risk representation forms are expressed in discrete time and space [29,30].

The objective of this study was to characterize the spatiotemporal trend of regional risk arising from soil and groundwater contamination. The USEPA risk assessment method and a numerical model were used to make a health risk assessment of oil tank leakage at a service station in Yibin, China. The study provides a practical and theoretical basis for protecting groundwater drinking water sources.

## 2. Study Area

The study area was located in the low mountains and hills in southern Sichuan (104°57′40″–105°14′33″ E, 28°22′20″–28°56′45″ N). This area belongs to the humid climate zone of the central subtropical Sichuan basin, and has an annual average temperature of 18.1 °C, annual average humidity of 83%, annual average sunshine duration of 1199.3 h, and an annual frost-free period of 347 d. According to long-term rainfall data, the annual average rainfall in the region is 1111 mm and varies obviously by seasons, in which 68% of the annual rainfall occurs from June to September, and only 2.3% occurs from December to February.

The service station examined in this study was located in the western suburbs of Yibin County in Sichuan Province, and covered an area of 3900 m^2^. The functional area of the service station mainly included the oil tank area and the oil filling work area. The oil tank area (110 m^2^) was distributed along the northern and western sides of the property. In the tank area, there were four buried single-wall oil tanks, each containing 30 m^3^. Residents lived within 500 m around the service station and were not served by a municipal water supply system. Thus, the residents used the groundwater as their source of drinking water and extracted the groundwater directly without purification. The pumping volumes ranged from 2.5 to 4 m^3^ d^−1^ (Figure 1). Since the gas station was put into operation in 2015, there have been many tank leaks and flooding overflows of the single-wall oil tanks. Pollutants that are characteristic of petroleum products have been detected in several monitoring wells at the study site. Among the pollutants, benzene, ethylbenzene, dichloroethane, and other organic contaminants are carcinogenic risk factors that threaten the safety of residents’ drinking water source.

According to the field investigation, the aquifer from which water is withdrawn by residential wells, which is polluted by contaminants leaked from the service station, is a shallow weathered fissured aquifer (J_2_s, Figure 1). To construct the hydrogeological model of the study area, exploratory drilling and other physical explorations were used to obtain the structural information about the aeration zone (i.e., vadose zone) and the aquifer. The hydrogeological information for the service station location is shown in Figure 2. The depth of exploration and drilling was 15–30 m, and the depth of the strongly weathered layer was 8.9–16.6 m. The strongly weathered layer was underlain by a moderately weathered layer. The drilling did not pass through the moderately weathered layer. The survey results indicated that the aeration zone varied from 1.1 m to 6.59 m in thickness, and was composed of loose sediments. Furthermore, the strongly weathered zone of bedrock had a vertical permeability coefficient in the range 0.5–1 m d^−1^. Otherwise, the permeability coefficient of the strongly weathered bedrock layer in main area of groundwater occurrence was in the range 1–8 m d^−1^, and the permeability coefficient of the moderately weathered layer was in the range 0.001–0.1 m d^−1^. Thus, the moderately weathered layer acted as a relative water barrier.

## 3. Materials and Methods

### 3.1. Sampling and Measurement Techniques

Taking the location of the oil depot as a reference point, soil samples and groundwater samples in the reference area and the potentially contaminated areas were collected during July 2019 (Table 1). Soil samples were collected according to the Technical Guidelines for Soil Sampling in China [31]. At sampling points 1#–6# in the potentially contaminated areas, 18 non-disturbed soil samples were collected using a power probe; three intact samples were retrieved at each location from depths 0.5–1 m, 1–2 m, and 2–3 m, respectively. Sampling point #7 was the background (reference) monitoring point, and was located 350 m north of the oil depot. A composite soil sample representing the 0.5–3 m soil layer was collected at sampling point #7. According to the composition characteristics of petroleum-related pollution sources, 20 heavy metals, inorganic substances, and organic contaminants posing a high threat to human health were selected for monitoring. Samples of soil were pretreated according to monitoring specifications [32], and placed in brown glass bottles fitted with tightly screwed caps. The samples were prepared for analysis by air drying, rough grinding, and fine grinding and then placed in different containers based on the requirements of analytical methods. At the same time, 19 water level measurements were made and 11 groundwater samples were collected. Groundwater samples were analyzed for the same parameters, as were soil samples. Samples were analyzed using standard methods of the American Public Health Association [33] to determine pollutant concentrations. Table 2 shows the detection method and minimum detectable value of each parameter. All water quality parameters are expressed in milligrams per liter (mg L^−1^).

### 3.2. Groundwater Modeling

The commonly used groundwater pollution transportation simulation software, Visual Modflow (https://www.waterloohydrogeologic.com/) was selected for application in this study. Visual Modflow predicts the impact of different management actions on pollutant transport under different boundary conditions in different saturated heterogeneous regions [34]. Visual Modflow establishes a groundwater flow model by applying a module to perform three-dimensional finite-difference numerical simulation of groundwater through a porous medium. This module involves the following partial differential equation [35]:(1)$$\frac{\partial }{\partial x_{i}}\left(K_{xx}\frac{\partial h}{\partial x}\right)+\frac{\partial }{\partial y}\left(K_{yy}\frac{\partial h}{\partial y}\right)+\left(K_{zz}\frac{\partial h}{\partial z}\right)-W=S_{s}\frac{\partial h}{\partial t}$$
where *h* is the piezometric head (L), *W* is the volumetric flux per unit volume that is represented for pumping, recharge or other sources, such as reservoirs (T^−1^), *S_s_* is the specific storage coefficient of the porous material (L^−1^), *t* is the time (T), *K_xx_*, *K_yy_*, and *K_zz_* represent the values of hydraulic conductivity along the *x*, *y*, and *z* coordinate axes, respectively (L T^−1^), and *x*, *y*, and *z* are the coordinate directions (L).

The MT3DMS (three-dimensional modular pollutant transport model, https://hydro.geo.ua.edu/mt3d/mt3dms2.htm) model is used widely to simulate solute transport in polluted aquifers. To determine the pollution plume, the transport model (MT3DMS) was used in combination with the flow field generated by the common flow model (Visual Modflow). In general, the mathematical model describing solute transport is as follows [36,37,38]:(2)$$\frac{\partial \left(\omega C^{k}\right)}{\partial t}=\frac{\partial }{\partial x_{i}}\left(\omega D_{ij}\frac{\partial C^{k}}{\partial x_{j}}\right)-\frac{\partial }{\partial x_{i}}\left(\omega v_{i}C^{k}\right)+q_{s}C_{s}{}^{k}+\sum R_{n}$$
where *C^k^* denotes *k* concentrations in water (M L^−3^), *ω* is the porosity of the porous medium (dimensionless), *t* is time (T), *x_i_* is the distance along the respective cartesian coordinate axis (L), *D_ij_* is the hydrodynamic dispersion cofficient (L^2^ T^−1^), *v_i_* is the seepage or linear porewater velocity (LT^−1^), *q_s_* is the volumetric flux of water per unit volume of aquifer representing sources (positive) and sinks (negative) (T^−1^), *C_s_* is the concentration of sources or sinks (M L^−3^), and ∑ *R_n_* is a chemical reaction term (M L^−3^ T^−1^).

### 3.3. Health Risk Assessment

Based on the “four-step method” proposed by the US National Academy of Sciences (Figure 3), the USEPA issued a series of technical documents and guidelines on risk assessment, and created a scientific risk assessment system. At present, some environmental risk assessment models in China and elsewhere consider air transmission and soil migration as pollutant pathways, and some only consider groundwater migration. Groundwater transmission is often considered in groundwater ecological vulnerability assessment and groundwater health risk assessment [39,40]. As outlined in Figure 3, the research process includes spot investigation, data collection, virulence assessment, exposure assessment, model construction, and risk characterization of pollution. Based on the “World Health Organization International Agency for Research on Cancer Carcinogens List” and the characteristics of pollutants from service stations, pollutants examined in this study were classified as carcinogens and non-carcinogens. Then, the carcinogenic slope factor (*SF*) and drinking water quality requirements were collected to serve as threshold values (R*f*D) for health risk assessment [41,42]. Grey correlation analysis was used to show that the soil pollution occurred near the oil tank, and the indicator of groundwater pollution was also detected in the groundwater that served as the drinking water of residents. Therefore, the exposure path for the risk assessment was assumed to be the ingestion of pollutants by residents drinking groundwater, and the risk values of carcinogens and non-carcinogens were expressed for oral absorption [39,40]:(3)$${\mathrm{CDI}}_{\text{oral-water}}=\frac{C_{w}\times IR\times EF\times ED}{BW\times AT}$$
where CDI_oral-water_ is the long-term daily exposure, *C_w_* is the concentration of a particular pollutant in groundwater (mg/(L·d)), *IR* is the daily water consumption (L/d), *EF* is the frequency, number of days exposed in a year (d/a), *ED* is the total years of exposure (a), *BW* is the weight of an adult (kg), and *AT* is the average exposure time (d). The values of the parameters are shown in Table 3.

Trace carcinogens can have adverse effects on human health. Researchers often use relevant carcinogenic risk values to express the risk of adverse effects. The equation for calculating low-dose carcinogenic risk (*Ro*) is as follows [1/mg·(kg·d)^−1^] [43]:(4)$${Ro=\mathrm{CDI}}_{\text{oral-water}}\times SF$$

The numerical value of total carcinogenic risk (*TRo*) is the sum of carcinogenic risk values of different substances, without considering their synergistic or antagonistic effects:(5)$${TR}_{O}=\sum_{i=1}^{n}Ro_{i}$$

The USEPA-recommended value is usually used to judge the risk of carcinogenesis. When *TRo* is less than 10^−6^, the risk of cancer is considered to be relatively low; when *TRo* is 10^−4^–10^−6^, cancer risk is considered likely to result in cancer; when *TRo* exceeds 10^−4^, the risk of cancer is relatively high [44,45,46].

The equation for calculating non-carcinogenic risk (*HI*) is:(6)$$HI=\frac{{\mathrm{CDI}}_{\text{oral-water}}}{RfD}$$

If *HI* ≤ 1, the exposure dose is lower than the threshold that causes adverse reactions, and the exposure is not expected to cause significant harm to health. If *HI* > 1, the exposure dose exceeds the threshold, and exposure may exert adverse non-carcinogenic effects on humans [45]. According to the above equations, the parameters are calculated in which the main variable is *C_w_* (the concentration of a particular pollutant in groundwater). When the instantaneous concentration of a pollutant is obtained from monitoring, the health risk can be calculated, but the results are instantaneous and discrete. In this study, numerical simulation was used to carry out health risk assessment and realize the continuous representation of groundwater carcinogenic risk value *TRo* and risk index *HI* in the time and space dimensions, which is conducive to risk zoning and health risk prevention and management of groundwater drinking water sources.

## 4. Results and Discussion

### 4.1. Status of soil Pollution and Identification of Pollution Factors

#### 4.1.1. Unsaturated Soil Pollution Characteristics

According to USEPA regional screening [47] levels and Ministry of Ecology and Environmental of China soil environmental quality [48,49], a single factor index was used. The maximum standard percentage of heavy metal constituents in each sample ranged between 0.001 and 0.529, and the peak concentrations of copper, lead, arsenic, hexavalent chromium, and manganese among the nine heavy metals monitored all appeared in #1-1 and #1-2 at monitoring point #1, which was located within the oil depot (Figure 1 and Table 1). Among the inorganic pollutants the fluoride concentration was 580.3–634.0 mg kg^−1^, accounting for 0.187–0.205 of the exceeding the standard rate. Cyanide was not detected in any samples. The maximum standard percentage of the nine organic pollutants monitored was 0.008–0.432. However, such organic pollutants were only detected at monitoring point #1 where the peak concentrations all appeared in #1-1. The concentrations of organic pollutants in #1-1, #1-2, and #1-3 at monitoring point #1 exhibited an obvious trend of decreasing with increasing depth. Monitoring data showed that the vertical infiltration of pollutants after oil tank leakage was the main pollutant pathway through the soil in the unsaturated zone underlying the oil tank. In the identification of groundwater pollution paths, the unsaturated zone is considered to be an important protective barrier for the groundwater environment because the vadose zone has the ability to intercept pollutants and prevent them from permeating. The vertical attenuation of pollutants at monitoring point 1# verifies this function. Analyzing the horizontal distribution of pollutants in soil, any difference between monitoring point #1 and other monitoring points should be affected by the presence of exogenous pollutants rather than by lateral movement of pollutants from the oil depot.

#### 4.1.2. Grey Relational Analysis

Grey relational analysis (GRA) was conducted to further analyze the pollution characteristics of oil tank leakage into the soil environment in the aeration zone, and to identify characteristic factors for the risk assessment of the groundwater environment. GRA is a multi-factor statistical analysis method that uses grey correlation degree to describe the strength and magnitude of the relationship between factors based on the sample data of each factor. If the sample data column indicates that the changes of factors at two different locations are basically the same, the correlation degree between them is large. On the contrary, the degree of correlation is small if the factors at the two locations do not change in a similar way. GRA has been applied in hydrogeological research fields, such as to study mining tunnels and sources of water in-rush from tunnels [50,51,52]. Thus, in this study GRA was used to quantify the correlation between the pollutants found at different monitoring points (#1–6) and those found at the background monitoring point (#7) based on the monitoring of all pollutants and the criterion of only examining the different types of pollutants.

##### Analysis Steps

Assume that *X*_0_ (soil background value) is a reference sequence:$$X_{0}=\left\{x_{0}(1),x_{0}(2),\cdots ,x_{0}(n)\right\}$$

*X*_1_, *X*_2_, …, *X*_*m*_ (sample detection values) are comparative sequences:$$X_{1}=\left\{x_{1}(1),x_{1}(2),\cdots ,x_{1}(n)\right\}\;X_{2}=\left\{x_{2}(1),x_{2}(2),\cdots ,x_{2}(n)\right\}$$
…
$$X_{m}=\left\{x_{m}(1),x_{m}(2),\cdots ,x_{m}(n)\right\}$$

The steps of GRA are (a) calculating the grey relevance coefficient, (b) calculating the grey relevance degree, and (c) grey relational analysis.

(a) Grey relevance coefficient *ξ*_0*i*_(*k*) can be calculated using Equation (7):(7)$$\xi_{oi}(k)=\frac{\begin{array}{c} \min_{i}\min_{k}\Delta_{i}(k)+\rho \max_{i}\max_{k}\Delta_{i}(k) \end{array}}{\begin{array}{c} \Delta_{i}(k)+\rho \max_{i}\max_{k}\Delta_{i}(k) \end{array}}$$
where, Δ*_i_*(*k*) = |*x*_0_(*k*) − *x_i_*(*k*)|; *k* = 1, 2, …, *n*; *i* = 1, 2, …, m, and ρ is the resolution coefficient, the general value for which is 0.5 [51].

(b) Grey relevance degree *γ*_0*i*_ can be calculated using Equation (8):(8)$$\gamma_{oi}=\frac{1}{N}\sum_{k=1}^{N}w_{k}\xi_{oi}(k)$$
where, $w_{k}$
(k=1,2, …, n) is the weight of each index. The coefficient of variation objective weighting method is used as defined by Equations (9)–(11) [50].

First, the standard deviation of each index is calculated ($\sigma_{k}$), which reflects the absolute variation degree of each index:(9)$$\sigma_{k}=\sqrt{\frac{1}{m}\sum_{i=1}^{m}{\left(x_{ik}-\bar{x_{k}}\right)}^{2}}(k=1,2,\ldots ,n)$$
where $\bar{x_{k}}$ is the average of the *k* indexes.

Next, the coefficient of variation of each index is calculated (*C_k_*), which reflects the relative variation degree of each index:(10)$$c_{k}=\sigma_{k}/\bar{x},\;(k=1,2,\ldots ,n)$$

Last, the weight of each index is obtained by normalizing the coefficient of variation of each index (*w_k_*):(11)$$w_{k}=c_{k}/\sum_{k=1}^{n}c_{k},\;(k=1,2,\ldots ,n)$$

The weight vector of each index is described as *W* = (*w*_1_, *w*_2_, …, *w*_n_).

(c) In the grey relational analysis, the degree of correlation *γ*_0*i*_ can be regarded as a measure of the overall correlation between the reference sequence and the comparison sequence. The closer *γ*_0*i*_ is to the value of 1, the better is the correlation between the reference sequence and the comparison sequence. It is generally believed that when *ρ* = 0.5, *γ*_0*i*_ ≥ 0.85 indicates a good correlation. Similarly, *γ*_0*i*_ = 0.7–0.85 indicates good correlation, *γ*_0*i*_ = 0.6–0.7 indicates acceptable correlation, *γ*_0*i*_ = 0.5–0.6 indicates poor correlation, and *γ*_0*i*_ < 0.5 indicates non-significant correlation [53].

##### Grey Correlation Degree Results

The grey correlation degree results for different types of pollutants are shown in Table 4. The correlation degree between the types of pollutants for each of the three sampling sections at monitoring point #1 and the background (#7) was in the range 0.375–0.567, indicating that the correlation varied from barely significant to insignificant; however, the correlation between sampling sections and background increased with the increase of sampling depth. In other words, the composition of soil at monitoring point #1 was significantly different from that at the reference point #7. The correlation degree between the types of pollutants for each sampling section at other monitoring points and the background monitoring point (#7) was excellent (0.950–0.990). Furthermore, the vertical correlation degree at these points exhibited no obvious pattern. Thus, GRA indicated that the oil tank leakage mainly polluted the underlying soil, and that horizontal migration range of the examined contaminants in the soil was relatively limited. Soil intercepts pollutants during the infiltration process, and retards or degrades some of them; thus, the influence of surface pollution sources on monitoring points deep in the soil profile is relatively weak, at least initially. However, this interception effect is limited. For example, the monitoring data from layers #1-1 and #1-2 and their correlation with background values are similar. These results suggest that the soil interception at the depth of 0–2 m had become saturated at the time of sampling. Therefore, the difference in pollutant concentrations and correlation between the two sections was no longer obvious.

Assuming that only one type of pollution is considered (i.e., ignoring interactions among pollutants), the GRA showed that for inorganic factors, the correlation degree between all samples and background values was high (0.989–1.000). Furthermore, fluoride and cyanide may not be considered as indicators of groundwater pollution. Among the remaining two pollutants, excluding organic pollutants and only considering the heavy metal pollutant, the correlation degree of monitoring point #1 with the background (#7) increased to 0.404–0.626; however, these correlations were barely significant or insignificant. The correlation degree of soil sections at other monitoring points was high (0.780–0.958). Considering only organic pollutants, the correlation degree of monitoring point #1 decreased to 0.333–0.520, but the correlation degree for soil layers at other monitoring points was high (as much as 1.000). In summary, GRA indicated that the oil tank leakage caused the most significant changes to the organic components in the soil, and the order of pollution significance was organic pollutant > heavy metal pollutant > inorganic pollutant. Therefore, the organic pollutant was selected as the indicator for the subsequent risk analysis. In analyzing the groundwater detection data, total petroleum hydro-carbon (THP) was used as the comprehensive indicator of organic pollutants for the non-carcinogenic risk assessment. Benzene, ethylbenzene, and methylene chloride were used as indicators in the carcinogenic risk assessment factors due to their carcinogenicity, with slope factors of 0.55, 0.011, and 0.002, respectively [41,42].

### 4.2. Risk Assessment

To determine the *C_w_* trend in the continuous space and time dimensions, the Visual Modflow software was used to construct a hydrogeological numerical model of the study area. Based on the monitored groundwater level and quality, the rationality of the model was verified. Thereafter, the trend of groundwater pollutant concentration was calculated. Finally, the numerical model was used to predict changes in *C_w_* and accurately depict the range of various risk areas in different periods.

#### 4.2.1. Model Construction and Calibration

##### Model Grid Construction and Parameterization

Based on hydrogeological drilling and physical exploration data, a groundwater model with an area of 0.51 km^2^ was constructed, in which the Yijiang River was defined as the excretory surface. The vertical cross-section of the simulated area was divided into a strongly weathered layer and a moderately weathered layer according to the permeability coefficients, and the depth of the strongly weathered layer was 15–25 m. Rectangular grids of 5 m × 5 m were used to subdivide the modeled area. The main factors controlling the accuracy of the model predictions were hydrogeological parameters and solute transport parameters, which were obtained by in situ hydrogeological experiments and other data collection techniques. The values of various parameters for the reasonable-fit model are shown in Table 5, including the values of hydraulic conductivities (K) of each layer. The model boundaries and other simulation features are shown in Figure 4.

##### Boundary Conditions and Pollution Source Setting

Boundary conditions reflect the process and intensity of a model and the external water quantity and material exchange, all of which are necessary for an accurate simulation [54]. The boundary conditions should be based on a correct understanding of the hydrogeological conditions in the simulated area, while ensuring a true reflection of the mathematical model. The hydrodynamic conditions of shallow weathered fissure water in China’s southwestern red bed aquifer are characterized by local topography and adjacent water bodies. Therefore, the boundary settings were as follows. The west side and the north side of the simulation area were upstream (up-gradient) of the simulation area and were set as the inflow boundaries. The east side was set as the outflow boundary in the lateral direction. The south side was the drainage boundary of the simulation area; combined with the hydrological and hydrodynamic conditions, this was set as the river boundary. The south of the river boundary was divided into different hydrogeological units and was set as the invalid unit. According to the actual groundwater survey, 14 water quality monitoring wells were distributed in the simulated area, and the water intake was 2.5–4 m^3^ d^−1^. At the same time, 19 water level monitoring wells were set for the verification of the water flow model; 11 water quality monitoring wells were set for the calibration of the solute transport model. The model domain (including the boundary conditions, positions of the observation wells and pumping wells) is shown in Figure 4.

##### Flow Model Calibration

The hydrogeological parameters measured in the survey, as well as the parameters of hydraulic conductivity, recharge, and specific yield, were set in the numerical model. The predicted output flow field was compared with the measured water level data to verify the rationality of the model [55]. The smaller was the value of the standard error of the estimate, the closer were the model predictions to the actual observation values, and the higher was the model accuracy. As shown in Figure 5, 19 observed wells were used in flow model validation. Table 6 shows the observed and predicted water levels (heads) for the various monitoring wells. Figure 5 shows that the standard error of the estimate was very small, only 0.109 m. Thus, the groundwater model predictions were consistent with the observed water heads.

After calibrating the flow model, the solute transport model was calibrated by trial-and-error [56]. According to the mathematical equation of solute transport (Equation (2)), the uncertain parameters affecting the solute transport model include the dispersion coefficient of pollutants and the pollutant flux; the pollutant flux is equal to the product of solvent infiltration and pollutant concentration. The dispersion coefficient is regarded as being quantified after the flow model has been successfully calibrated. The only parameters affecting the predicted concentration of a pollutant are solvent infiltration and initial pollutant concentration. By adjusting the pollutant flux, the model predictions and the observed values of the pollutant factors were compared.

Figure 6 shows that data from 11 water quality observation wells were used for the transport model calibration. After calibrating the solute-transport model, the root mean square error for THP was 1.416 mg L^−1^. The absolute residual mean was 0.853 mg L^−1^.

The model simulations showed that leakage from the oil depot affected the groundwater environment around the oil depot and down-gradient in the direction of groundwater flow. The maximum THP concentration (25 mg L^−1^) was predicted at the location of the pollution source; this corresponded to a risk assessment value (*HI*) of 26, which was much larger than the risk standard value of 1. Furthermore, the predicted pollution coverage was 25,580 m^2^, which exceeded the footprint of the service station and extended 149 m down-gradient of the southeast side of the station.

#### 4.2.2. Analysis of Model Simulation Results

In reality, once pollution sources are interrupted and other remediation measures are taken, the maximum pollutant concentration in groundwater typically continues to increase, the migration range of pollutants keeps expanding, and the health risk gradually accumulates. The groundwater model was used to predict the *HI* and *TRo* characteristics in five and 10 years (Figure 7).

After five years, the predicted maximum non-carcinogenic risk index *HI* was 25.2, and the area in which *HI* exceeded the value of 1 was 29,593 m^2^; this area extended 193.0 m down-gradient from the southeast boundary of the service station and was only 6.9–46.2 m from the surrounding intake wells for residents. Compared with the situation at the time of the survey, the predicted risk area increased by 16.22% after five years. Similarly, the predicted maximum value of the carcinogenic risk index *TRo* was 4.73 × 10^−5^ after five years, and the area in which *TRo* exceeded 10^−6^ was 40,093 m^2^. The risk area extended 222.4 m down-gradient from the southeast boundary of the service station and was 18.5–60.5 m from the surrounding residential area.

After 10 years, the predicted maximum non-carcinogenic risk index *HI* was 25.3, and the area in which *HI* exceeded the value of 1 was 39,873 m^2^, which extended 255.4 m down-gradient from the southeast boundary of the service station and was 5.9–46.2 m from the surrounding residential area. Compared with the situation at the time of the survey, the risk area increased by 56.60%. Similarly, the predicted maximum value of the carcinogenic risk index *TRo* was 4.8 × 10^−5^, and the area in which *TRo* exceeded 10^−6^ was 53,488 m^2^. The risk area extended 285.9 m down-gradient from the southeast boundary of the gas station and was 22.7–65.2 m from the surrounding residential area.

The model simulations predicted that the maximum pollutant concentrations in groundwater would increase in 5–10 years, but at a slow rate (1.2–1.5% per annum). Likewise, the average expansion rate of the total area of the risk was predicted to be 0.033–0.055 m^2^ per annum. The maximum risk distance along the main groundwater flow direction was predicted to increase by 0.057–0.071 m per annum. Conversely, the pollutant diffusion distance up-gradient from the service station was predicted to increase very little in 10 years, while the vertical diffusion distance in the main flow path of groundwater was predicted to increase at a rate of 0.015 m per annum. Thus, drinking water wells located down-gradient of the main volume of polluted groundwater were most at risk and should be a priority for pollution prevention and control efforts. According to the results of the field investigation, drinking water wells W18 and W19 were already contaminated at the time of the survey. After 10 years, the *TRo* and *HI* values of W19 were predicted to be 1.6 × 10^−6^ and 1.3, respectively. Thus, urgent action is needed to implement measures to improve the soil and groundwater environment.

## 5. Conclusions and Recommendations

This study examined the soil and groundwater around a service station in Yibin, China, which was the site of a leaking underground oil storage tank. Combining numerical modeling and health risk assessment, we established a hydrogeological numerical model of the study area using Visual MODFLOW software (Waterloo Hydrogeologic, Vancouver, Canada) to determine the health risks trend in the continuous space and time dimensions. The results justify the following conclusions. Pollution of the soil by leakage from the buried oil tank is mainly concentrated in the oil storage area. The pollutants leaked from the oil tank also have penetrated groundwater in the vicinity of the station. The groundwater is the source of drinking water for local residents. According to the risk assessment that was performed, the contamination in this risk zone may harm human health and even cause cancer. According to simulation results, the contamination will worsen over time. Therefore, pollution control measures must be taken to reduce the risk posed to human health.

Among the measures that can be taken, replacing the existing single-wall storage tanks with double-walled tanks manufactured using upgraded material should be part of any prevention and control program to eliminate the source of pollution at the station. Furthermore, contaminated soil can be removed, but this may cause “secondary pollution”. Implementing strict anti-seepage measures in the storage tank area should be considered to create artificial barriers for the pollutant migration path. Moreover, to promote the rehabilitation speed of contaminated groundwater, taking measures such as controlling the local hydrodynamic conditions in the aquifer and setting grouting curtains in the down-gradient direction of pollutant migration can help mitigate pollutant movement and increase the groundwater flow flux. The polluted groundwater can be removed and treated or disposed. In summary, a variety of source-process-end controls are needed to eliminate the health risks of drinking contaminated groundwater at the study site and effectively ensuring the future safety this drinking water source.

## Figures and Tables

**Figure 1 ijerph-16-03245-f001:**
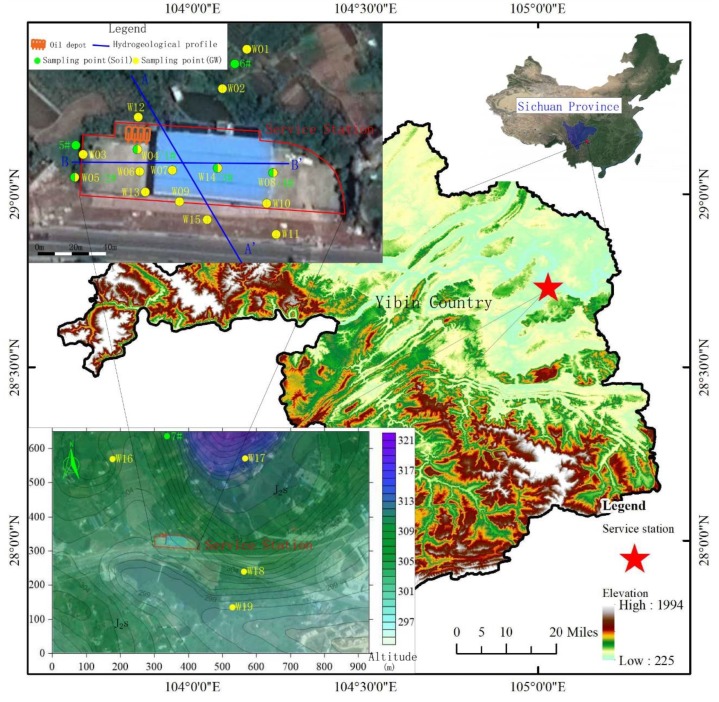
Location of the study area in Sichuan Province, China. The detailed inset in the upper left shows the service station layout and various monitoring points. The locations of monitoring wells are shown in the inset in the lower left.

**Figure 2 ijerph-16-03245-f002:**
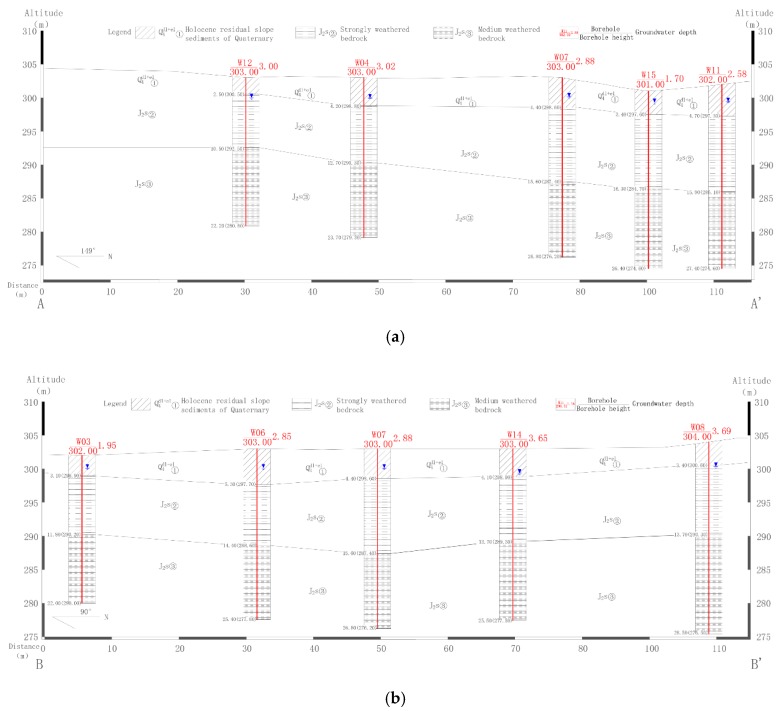
The geophysical exploration profiles of transects: (**a**) profile A-A’, (**b**) profile B-B’ (see Figure 1).

**Figure 3 ijerph-16-03245-f003:**
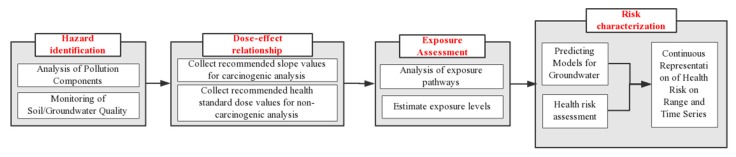
Process of health risk assessment.

**Figure 4 ijerph-16-03245-f004:**
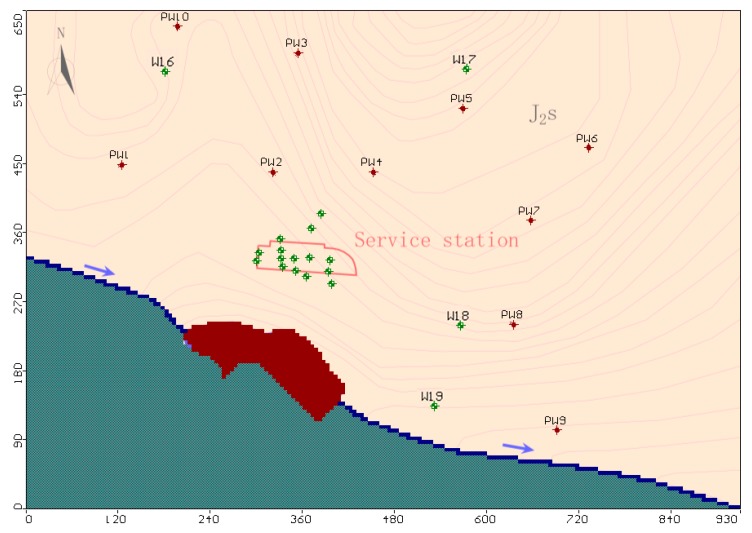
Positions of boundary, tracer particle, and observation wells of the study area.

**Figure 5 ijerph-16-03245-f005:**
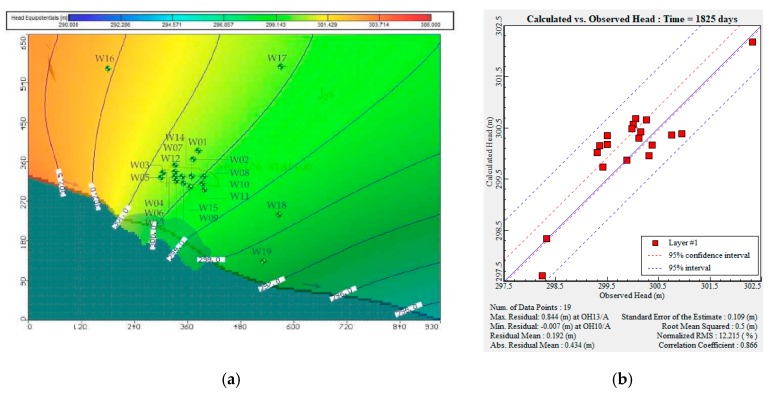
Calibration of initial flow field of groundwater: (**a**) schematic chart of initial flow field of groundwater, which indicates the direction and magnitude of groundwater flow; (**b**) residual analysis diagram of groundwater level in the research area. Transport model calibration.

**Figure 6 ijerph-16-03245-f006:**
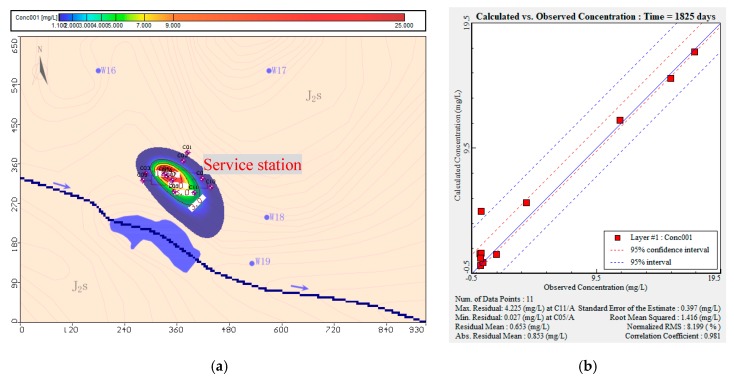
Calibration of initial pollution concentration of total petroleum hydro-carbon (THP): (**a**) predicted pollutant concentration distribution after calibration of the model; (**b**) residual analysis of predicted and detected THP concentrations in the study area.

**Figure 7 ijerph-16-03245-f007:**
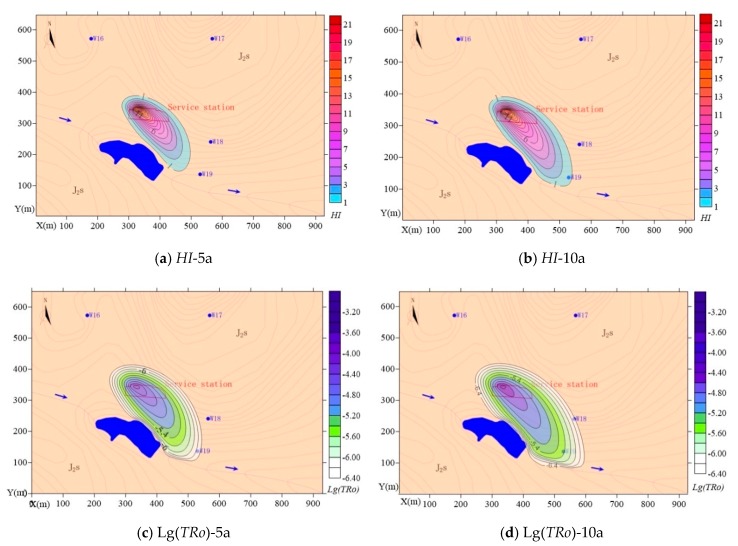
Predicted health risks: (**a**) spatial distribution of non-carcinogenic risk after five years; (**b**) spatial distribution of non-carcinogenic risk after 10 years; (**c**) spatial distribution of carcinogenic risk after five years; (**d**) spatial distribution of carcinogenic risk after 10 years.

**Table 1 ijerph-16-03245-t001:** Details of groundwater sampling.

Soil Samples			Groundwater Samples			
ID		Location	ID	Location	Monitoring Elements	Groundwater Level
1#	1-1#	Next to the south of the oil depot (W04)	W01	65 m to the northeast of the oil depot	Head and Quality	300.95 m
	1-2#		W02	40 m to the northeast of the oil depot		300.76 m
	1-3#		W03	22 m to the southwest of the oil depot		299.7 m
2#	2-1#	31 m to the southwest of the oil depot (W5)	W04	Next to the south of the oil depot		299.5 m
	2-2#		W05	31 m to the southwest of the oil depot		299.55 m
	2-3#		W06	12 m to the south of the oil depot		300.15 m
3#	3-1#	36 m to the southeast of the oil depot (W14)	W07	18 m to the southeast of the oil depot		300.12 m
	3-2#		W08	62 m to the southeast of the oil depot		300.31 m
	3-3#		W09	32 m to the southeast of the oil depot		299.5 m
4#	4-1#	62 m to the southeast of the oil depot (W08)	W10	66 m to the southeast of the oil depot		299.88 m
	4-2#		W11	77 m to the southeast of the oil depot		299.42 m
	4-3#		W12	5 m to the north of the oil depot		300 m
5#	5-1#	30 m to the west of the oil depot	W13	24 m to the south of the oil depot	Head	299.5 m
	5-2#		W14	36 m to the southeast of the oil depot		299.35 m
	5-3#		W15	47 m to the southeast of the oil depot		299.3 m
6#	6-1#	56 m to the northeast of the oil depot	W16	272 m to the northwest of the oil depot		302.33 m
	6-2#		W17	310 m to the northeast of the oil depot		300.37 m
	6-3#		W18	246 m to the southeast of the oil depot		298.32 m
7#	Background point	350 m to the north of the oil depot	W19	279 m to the southeast of the oil depot		297.5 m

**Table 2 ijerph-16-03245-t002:** Details of analytical methods and minimum detection values.

Number	Monitoring Factors		Soil			Groundwater		
			Detection Method	Instrument	Minimum Detectable Value (mg/L)	Detection Method	Instrument	Minimum Detectable Value (mg/L)
1	As		Inductively Coupled Plasma Mass Spectrometry (ICP-MS)	Inductively Coupled Plasma Mass Spectrometer: Nex ION 350X	0.00009	Metal Index Atomic Fluorescence Spectrometry	Atomic fluorescence spectrophotometer: AFS-930	0.01
2	Cu				0.00009	Flame Atomic Absorption Spectrophotometry	Atomic Absorption Spectrophotometer AA-7000	1
3	Zn				0.0008			0.5
4	Mn				0.00006			0.1
5	Ni				0.00007			5
6	Pb				0.00007	Graphite Furnace Atomic Absorption Spectrophotometry (GF-AAS)	Atomic Absorption Spectrophotometer AA-9000T	0.1
7	Cd				0.00007			0.01
8	Hg		Metal Index Atomic Fluorescence Spectrometry	Atomic fluorescence spectrophotometer: AFS-930	0.00001	Cold Atomic Absorption Spectrophotometry	Differential mercury analyzer (WCG209)	0.1
9	Cr^6+^		Diphenyl carbonyl hydrazine spectrophotometry	Ultraviolet-visible Spectrophotometer: UV-7504	0.001	Diphenyl carbonyl hydrazine spectrophotometry	Ultraviolet-visible Spectrophotometer: UV-7504	0.16
10	Cyanide		Spectrophotometry method		0.004	Colorimetric method of isonicotinic acidpyrazolone	Ultraviolet-visible Spectrophotometer: UV-7504	0.04
11	Fluoride		Ion chromatography	Ion chromatography: ECOIC	0.006	Ion selective electrode analysis method	PHSJ-4A	5
12	Naphthalene		Purge and trap/gas chromatography-Mass spectrometry	Gas chromatography-mass spectrometer GC-MS: SHIMADZ QP-2010 Ultra	0.0001	Purge and trap/gas chromatography-mass spectrometry	Gas chromatography-mass spectrometer GC-MS: SHIMADZ QP-2010 Ultra	0.0004
13	Benzene				0.0001			0.0009
14	Methylbenzene				0.0001			0.0009
15	Ethylbenzene				0.0001			0.0009
16	m-Xylene; para-xylene				0.0002			0.0008
17	ortho-xylene				0.0002			0.0008
18	Dichloromethane				0.0003			0.0011
19	MTBE (methyl tert-butyl ether)		Determination of volatile organic compounds by purge/trap/gas chromatography mass spectrometry	Gas chromatograph-mass spectrometer: Agilent 7890A-5975C	0.0005	Determination of volatile organic compounds by purge/trap/gas chromatography mass spectrometry	Gas chromatograph-mass spectrometer: Agilent 7890A-5975C	/
20	Total petroleum hydro-carbon (TPH)	C6-C9	Nonhalogenated Organics Using GC/FID (Flame Ionization Detector)	Gas chromatograph: 7890B	0.05	Determination of Non-halogenated Organic Compounds by GC/FID	Gas Chromatograph (GC) 7890B	0.1
		C10-C36			0.03	Gas Chromatography	Gas Chromatograph GC-2010plus	0.43

**Table 3 ijerph-16-03245-t003:** Parameters of oral absorption pathway in health risk assessment.

Symbol	Name	Unit	Recommended Values (Adult)	Symbol	Name	Unit	Recommended Values (Adult)
*C_w_*	Concentration of *i* in groundwater	Mg/(L·d)	Measured value	*ED*	Total years of exposure	a	70–40
*IR*	Daily water consumption	L/d	2	*BW*	Weight	kg	61.52
*EF*	Exposure frequency, number of days exposed in a year	d/a	350	*AT*	Average exposure time	d	25,550–14,600

**Table 4 ijerph-16-03245-t004:** Grey correlation analysis results.

Sampling Point		Comprehensive Pollution Factor	Heavy Metal Pollution Factor	Inorganic Pollution Factor	Organic Pollution Factor
1#	1-1#	0.375	0.411	0.989	0.333
	1-2#	0.387	0.404	0.990	0.353
	1-3#	0.562	0.626	1.000	0.520
2#	2-1#	0.982	0.923	1.000	1.000
	2-2#	0.990	0.958	0.998	1.000
	2-3#	0.972	0.875	0.999	1.000
3#	3-1#	0.977	0.899	0.999	1.000
	3-2#	0.983	0.928	0.993	1.000
	3-3#	0.978	0.905	0.993	1.000
4#	4-1#	0.964	0.844	0.994	1.000
	4-2#	0.967	0.856	0.989	1.000
	4-3#	0.976	0.894	0.995	1.000
5#	5-1#	0.963	0.837	0.996	1.000
	5-2#	0.950	0.780	1.000	1.000
	5-3#	0.958	0.817	0.998	1.000
6#	6-1#	0.968	0.859	0.998	1.000
	6-2#	0.963	0.837	0.997	1.000
	6-3#	0.968	0.861	0.997	1.000

**Table 5 ijerph-16-03245-t005:** Model parameter values used in numerical simulations.

Model Parameter		Value
Length of model domain in *x*-direction/m		930 m
Length of model domain in *y*-direction/m		650 m
Dimension of one grid cell/m		5 × 5
Average annual rainfall		1110
Rainfall infiltration coefficient		0.08–0.12
Specific yield (Sy)		0.1
Effective porosity		0.1
Total porosity		0.15
Specific storage, (*S_s_*/m)		1.0 × 10^−7^
hydraulic conductivity (*K_x_*, *K_y_*)	First layer	0.0008
	Second layer	0.00001
*K_z_*		1/10*K_x_*
D_L_ longitudinal dispersion		0.467
D_T_ Ratio of longitudinal dispersion to transverse dispersion		1/10DL

**Table 6 ijerph-16-03245-t006:** Computed versus observed head values.

Well/Point Name	Obs. (Observation Value)	Calc. (Calculated Value)	Calc.-Obs.	Well/Point Name	Obs.	Calc.	Calc.-Obs.
W1	300.95	300.39	−0.56	W11	299.42	299.74	0.32
W2	300.76	301.36	0.6	W12	300.00	300.57	0.57
W3	300.05	299.68	−0.37	W13	299.50	300.34	0.84
W4	299.98	299.49	−0.49	W14	299.35	300.15	0.80
W5	300.26	299.65	−0.61	W15	299.30	300.01	0.71
W6	300.15	300.42	0.27	W16	302.33	302.18	−0.15
W7	300.12	300.30	0.18	W17	300.37	300.16	−0.21
W8	300.31	299.96	−0.35	W18	298.32	298.34	0.02
W9	299.50	300.18	0.68	W19	297.80	297.60	−0.20
W10	299.88	299.87	−0.01

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
