# Peer review of "Health Risk Assessment of Groundwater Contaminated by Oil Pollutants Based on Numerical Modeling"

_ijerph, 2019, doi:10.3390/ijerph16183245_

Round 1
Reviewer 1 Report
1. General Comments:
This paper investigated "the risk assessment of groundwater polluted by oil". It is an interesting topic but before considering for publication, it needs some improvements.
Water pollution has worsened in several areas around the world, particularly in areas with threats of high contamination. Therefore, risk assessment of polluted-waters has gained researchers' attentions.
2. Abstract:
2.1. Start the abstract with motivational statement.
2.2. Page 1, Line 14; "Gray correlation analysis shown that..." should be corrected to "Gray correlation analysis showed that..."
2.3. Write keywords alphabetically.
3. Introduction:
3.1. Quality of Figure 2 is not acceptable!
4. Materials and Methods:
4.1. Why did you take samples just during July? Usually for risk assessment, sampling should be done during wet and dry seasons!
Author Response
Dear reviewers,
Thank you for your comments on our article. We have finished the revisions and marked the changes in manuscript. We provided a list of the revisions. If have any questions, you can contact us at any time. At the same time, I feel admiration for your carefulness. Thank you very much for your attention and consideration.
Response:
Start the abstract with motivational statement.Replay: Thank you for your valuable comment. According to the revision suggestion, we rewrote the abstract, thus emphasizing the research purpose and the research methods. Now we can spell out that in order to accurately evaluate the time and scope of groundwater pollution hazards to human health, and then helps the decision-making process for remediation of polluted soil and groundwater in service stations, we have chosen the method of coupling numerical method with health risk assessment to conduct effective research.
Page 1, Line 14; "Gray correlation analysis shown that..." should be corrected to "Gray correlation analysis showed that....
Replay: Thank you for your suggestion. We corrected this sentence.
Write keywords alphabetically.
Replay: Thank you for your valuable comment, we changed the order of keywords.
Quality of Figure 2 is not acceptable!
Replay: Thank you for your kind reminding. We replaced the illustrations
Why did you take samples just during July? Usually for risk assessment, sampling should be done during wet and dry seasons!
Replay: Thanks for this reviewer’s good suggestion. Let me explain to you the reason why we determined the sampling time. First of all, we found oil tank leakage shortly before sampling. We timely sampled to determine the pollution status. And the monitoring results of soil and groundwater have shown the pollution status of the region. Secondly, in Sichuan, China, July is the rainy season. To sum up, we chose to collect samples in July.
With kind regards,
Kai Song
Reviewer 2 Report
The paper contains relevant information about risk assesment in oil facilities near inhabited sites.
The paper is well written and all sections are clear and well structured.
It is recommended to include in the conclusions some ideas about the methods used to perform the risk projections
Author Response
Dear reviewers,
Thank you for your comments on our article. We have finished the revisions and marked the changes in manuscript. We provided a list of the revisions. If have any questions, you can contact us at any time. At the same time, I feel admiration for your carefulness. Thank you very much for your attention and consideration.
Response:
It is recommended to include in the conclusions some ideas about the methods used to perform the risk projections.Replay: Thank you for your valuable comment. We added the idea of a risk prediction method to our conclusion.
With kind regards,
Kai Song
Reviewer 3 Report
Dear Authors
In my opinion the theme of the article is very actual and interesting for the readers of the journal.
This study examined the soil and groundwater around a service station in Yibin, China, which was the site of a leaking underground oil storage tank.
The authors used the health risk assessment method coupled numerical simulation to assess the health risk of a drinking water source contaminated by oil depot leakage.
It was collected and analyzed soil and groundwater samples for 20 pollutants.
This research shows that fifty-six percent of the heavy contaminants and 100% of the organic contaminants exhibited maximum values at the location of the oil depot. The correlation of environmental impact after oil depot leakage followed the order: organic pollutants > heavy metals > inorganic pollutants.
Pollution of the soil by leakage from the buried oil tank is mainly concentrated in the oil storage area.
The pollutants leaked from the oil tank also have penetrated groundwater in the vicinity of the station, being the groundwater is the source of drinking water for local residents.
According to the risk assessment that was performed, the contamination in this risk zone may harm human health and even cause cancer.
The authors warn for measures that can be taken, such as replacing the existing single-wall storage tanks with double-walled tanks manufactured using upgraded material should be part of any prevention and control program to eliminate the source of pollution at the station.
Contaminated soil may be removed from the local; however authors highlight that secondary pollution could be a consequence of this action.
The study concluded suggesting a variety of source-process-end controls that are needed to eliminate the health risks of drinking contaminated groundwater at the study site and that effectively ensuring the future safety this drinking water source. For example, promote the rehabilitation speed of contaminated groundwater, taking measures such as controlling the local hydrodynamic conditions in the aquifer and setting grouting curtains in the down-gradient direction of pollutant migration can help mitigate pollutant movement and increase the groundwater flow flux
The paper is well structured, the title and abstract clearly describe the content of the manuscript, and the language is correct and clear.
In my opinion this manuscript is ready to be accepted.
Best regards
Author Response
Dear reviewers,
Thank you for your comments on our article. We checked the details of the article again and made improvements.. We provided a list of the revisions. If have any questions, you can contact us at any time. At the same time, I feel admiration for your carefulness. Thank you very much for your attention and consideration.
Response:
We found that Figure 6 had some minor problems, so I revised it in the manuscript and added the PDF of Figure 6.
With kind regards,
Kai Song